# Deciphering the Molecular Mechanisms of Reactive Metabolite Formation in the Mechanism-Based Inactivation of Cytochrome p450 1B1 by 8-Methoxypsoralen and Assessing the Driving Effect of phe268

**DOI:** 10.3390/molecules29071433

**Published:** 2024-03-22

**Authors:** Emadeldin M. Kamel, Maha A. Alwaili, Hassan A. Rudayni, Ahmed A. Allam, Al Mokhtar Lamsabhi

**Affiliations:** 1Chemistry Department, Faculty of Science, Beni-Suef University, Beni-Suef 62514, Egypt; emad.abdelhameed@science.bsu.edu.eg; 2Department of Biology, College of Science, Princess Nourah bint Abdulrahman University, P.O. Box 84428, Riyadh 11671, Saudi Arabia; 3Department of Biology, College of Science, Imam Mohammad Ibn Saud Islamic University, Riyadh 11623, Saudi Arabia; harudayni@imamu.edu.sa (H.A.R.); ahmed.aliahmed@science.bsu.edu.eg (A.A.A.); 4Department of Zoology, Faculty of Science, Beni-Suef University, Beni-Suef 65211, Egypt; 5Departamento de Química, Módulo 13, Universidad Autónoma de Madrid, Campus de Excelencia UAM-CSIC Cantoblanco, 28049 Madrid, Spain; 6Institute for Advanced Research in Chemical Sciences (IAdChem), Universidad Autónoma de Madrid, 28049 Madrid, Spain

**Keywords:** cytochrome P450, DFT, 8-methoxypsoralen, catalyzed biotransformation, molecular docking, molecular dynamics simulations

## Abstract

This study provides a comprehensive computational exploration of the inhibitory activity and metabolic pathways of 8-methoxypsoralen (8-MP), a furocoumarin derivative used for treating various skin disorders, on cytochrome P450 (P450). Employing quantum chemical DFT calculations, molecular docking, and molecular dynamics (MD) simulations analyses, the biotransformation mechanisms and the active site binding profile of 8-MP in CYP1B1 were investigated. Three plausible inactivation mechanisms were minutely scrutinized. Further analysis explored the formation of reactive metabolites in subsequent P450 metabolic processes, including covalent adduct formation through nucleophilic addition to the epoxide, 8-MP epoxide hydrolysis, and non-CYP-catalyzed epoxide ring opening. Special attention was paid to the catalytic effect of residue Phe268 on the mechanism-based inactivation (MBI) of P450 by 8-MP. Energetic profiles and facilitating conditions revealed a slight preference for the C4′=C5′ epoxidation pathway, while recognizing a potential kinetic competition with the 8-OMe demethylation pathway due to comparable energy demands. The formation of covalent adducts via nucleophilic addition, particularly by phenylalanine, and the generation of potentially harmful reactive metabolites through autocatalyzed ring cleavage are likely to contribute significantly to P450 metabolism of 8-MP. Our findings highlight the key role of Phe268 in retaining 8-MP within the active site of CYP1B1, thereby facilitating initial oxygen addition transition states. This research offers crucial molecular-level insights that may guide the early stages of drug discovery and risk assessment related to the use of 8-MP.

## 1. Introduction

8-Methoxypsoralen (8-MP), also recognized as methoxsalen, is a naturally occurring phytochemical found in numerous plant species and classified under furocoumarins [1]. Psoralen–UVA (PUVA) photochemotherapy, employing 8-MP, is commonly used for treating various pervasive skin diseases [2]. Upon UV light exposure following oral administration, 8-MP exhibits photosensitizing effects [3]. Therefore, it has been clinically utilized in treating conditions like psoriasis, eczema, vitiligo, and certain types of cutaneous lymphomas [3]. Previous studies indicated that 8-MP can potently inhibit numerous isoforms of cytochrome P450 (P450) enzymes [4]. It was also demonstrated that pre-treatment with 8-MP notably mitigates lung tumorigenesis induced by certain xenobiotics in experimental animals [5]. This antitumor effect is likely due to the 8-MP ability to inhibit xenobiotic biotransformation catalyzed by deactivating P450 isozymes, particularly CYP2As, present in the lungs of experimental animals [5]. However, the precise mechanisms delineating the metabolic pathways through which 8-MP inactivates P450 remain underexplored.

A myriad of endogenous and exogenous substances, including certain FDA-approved anti-cancer drugs, undergo metabolism via NADPH-dependent Phase I by a specific P450 isozyme, known as CYP1B1 [6]. This isozyme has been identified by numerous researchers as an important target for a variety of reasons. Among these, is the predominant overexpression of CYP1B1 in extrahepatic tissues, such as those of ovarian, lung, brain, breast, and colon cancers, whereas its presence in adjacent normal tissues is scarce [7]. Furthermore, this isozyme rapidly biotransforms many anti-cancer drugs, thereby contributing to drug resistance [8]. Consequently, certain tumor cell lines may develop drug resistance as a result of the mechanism-based inactivation (MBI) by P450 1B1 [8]. CYP1B1 is reported to be the most prevalent isoform of P450 in the human epidermis, with substantial inter-individual variation in its intrinsic expression and regulation by UVR and PUVA [9]. These observations emphasize the importance of CYP1B1 as an intriguing target. Moreover, prior computational investigations identified active amino acid residues in the active site of CYP1B1, including aromatic residues Phe134, Phe231, and Phe268 [10]. These residues are known to preferentially engage in thermodynamic π–π stacking interactions with various aromatic substrates and recognized inhibitors [10].

MBI of P450s generally commences with the formation of reactive intermediates of a substrate, which can bind irreversibly to the enzyme’s active site [11,12,13,14]. This binding interaction renders the enzyme inactive, which can consequently alter drug efficacy [15]. However, the MBI of P450 serves as a critical method for examining the structure–function relationships of various P450 isozymes [15]. A notable drawback of MBI is its potential interference with P450’s fundamental metabolic function, potentially leading to undesirable drug–drug interactions or toxicity [16]. Despite this, the MBI of P450 can be advantageous as it increases the possibility of achieving a significant degree of selectivity when targeting a specific P450 isozyme [17]. P450 enzymes are characterized by their multifunctionality, facilitating the production of reactive metabolites and contributing to detoxification processes [11]. This intriguing functional versatility is primarily attributed to P450’s capability to incorporate a single oxygen atom into various P450-interacting drugs [12]. Our published data elaborate on the various stages involved in the MBI of P450 and the different types of inhibitors [11]. The utilization of P450 MBI has proven beneficial in identifying the heme porphyrin moiety within the active site of P450 and in determining the binding modes of various substrates [15]. Thus, studying these catalyzed biotransformation mechanisms can yield invaluable information for both the chemical and pharmaceutical industries.

Oral administration of various drugs often results in the liver being exposed to relatively high concentrations of these drugs and their reactive metabolites. The fundamental principle of drug biotransformation is to convert lipophilic drugs into hydrophilic reactive metabolites, facilitating their excretion [18]. However, some xenobiotics undergo biotransformation to form electrophilic species, a process known as bioactivation, which is often the initial step in many drug-induced toxicities [19]. These highly electrophilic reactive metabolites may alter the functional nature of biological macromolecules and enhance the covalent bonding to proteins [20]. These effects are believed to be linked to rare but serious idiosyncratic drug reactions and interactions [20]. Thus far, all studied instances of MBI involving various P450s have revealed at least one or more key reactive fragments that could be oxidatively metabolized by P450. Substrates without such key fragments are considered incapable of yielding reactive metabolites [21]. Therefore, it is crucial to identify the reactive metabolites in the MBI of P450 induced by various inhibitors. Such knowledge is highly beneficial for the early prediction of potential drug toxicities and for logical drug design and development to prevent idiosyncratic toxicity [21].

Quantum chemical density functional theory (DFT) calculations have offered invaluable insights into various pathways of P450-catalyzed biotransformation [22]. The current study is part of a continuing series that our team is conducting, investigating how diverse drugs, each with unique binding pathways, can inactivate P450 enzymes [11,12]. In this work, we utilized quantum chemical DFT calculations to examine the mechanistic details underlying the MBI of P450 by 8-MP. We offer an intensive discussion on the potential binding sites in 8-MP that are susceptible to P450 interaction. We also employed molecular docking analysis to evaluate the binding modes of CYP1B1 with 8-MP. As a result, we studied the influence of the key residue, Phe268, on the mechanism of 8-MP epoxidation catalyzed by P450. Furthermore, we thoroughly investigated the possible mechanisms of the formation of various reactive metabolites from the 8-MP-P450 catalyzed epoxidation products.

## 2. Results and Discussion

### 2.1. Molecular Docking Analysis

We conducted molecular docking to explore the binding profile and the biological environment implicated in the interaction between 8-MP and the active site of CYP1B1. This isozyme was chosen for the study due to its prevalence as the predominant isoform of P450 in the human epidermis [9], meaning that the 8-MP substrate will be in substantial direct contact with this isozyme during its therapeutic application for various skin diseases. The interaction representation between 8-MP and P450 1B1, the residues exhibiting hydrophobic interactions with 8-MP, and the surface representation of the 8-MP-P450 complex interaction are illustrated in Figure 1. Intriguingly, all interactions of 8-MP with the active site of CYP1B1 detected were exclusively hydrophobic, corroborating the previously reported hydrophobic character of the active site of numerous P450 isozymes [23]. This hydrophobic interaction implies that further hydrolysis of metabolic products is improbable. 8-MP was found to dock into the primary binding site of the enzyme with an intricate network of hydrophobic interacting residues. A notably low binding affinity (−9.6 kcal mol^−1^) was calculated for this complex, indicating the compatibility of 8-MP with the CYP1B1 binding cavity. Significant key residues Phe268, Phe23, and Phe134 were identified in the potential binding mechanism of this complex. These residues were shown to play substantial roles in the metabolic interactions of CYP1B1 [6]. Additionally, these phenylalanine residues can induce thermodynamically favorable π–π interactions involved in molecular recognition and conforming to the substrate in the protein binding site [11]. Other detected hydrophobic interacting residues like Ala330, Gly329, Asn228, and Asp326 were recognized as major contributors to the free-binding energy for many substrates [6]. Therefore, the molecular docking assessment results led to the conclusion of the activity of 8-MP against CYP1B1, affirming the potency of 8-MP as a P450 inhibitor.

### 2.2. Molecular Dynamic Simulations

To assess the stability of 8-MP within the binding site of P450, we conducted MD simulations to screen its interaction with the CYP1B1 isozyme. The complex with the lowest binding affinity from molecular docking analysis was employed for the MD simulation run. We extensively analyzed the 30 ns MD trajectories, focusing on parameters such as root mean square deviations (RMSD), root mean square fluctuations (RMSF), radius of gyration (Rg), solvent accessible surface area (SASA), interaction energies, and hydrogen bonding pattern for both CYP1B1 and the 8-MP-CYP1B1 complex. 

RMSD analysis elucidates structural variations occurring within the molecule throughout the course of molecular dynamics simulations. The RMSD plot illustrates these structural changes relative to their initial state at 0 ns through the simulation’s conclusion at 30 ns. As shown in Figure 2A, during the initial equilibration, the RMSD values for the enzyme and the 8-MP-P450 complex exhibited a rising trend, increasing from 0.12 to 0.2 nm over the first 7 ns. Then, both systems reached equilibrium and maintained within the range of approximately 0.15 to 0.22 nm until the end of the simulation with an average RMSD value of 0.17 ± 0.06 nm. Thus, these findings indicate that P450 and 8-MP-P450 systems maintained a stable trajectory throughout the 30 ns simulation without significant deviations. Furthermore, the RMSD analysis of the 8-MP revealed fluctuations within the standard range of 0.02–0.07 nm from 3 to 30 ns with an average RMSD value of 0.05 ± 0.005 nm, underscoring the ligand’s high stability within the binding cavity.

Next, we computed the time-averaged RMSF values of CYP1B1 residues in both the absence and presence of 8-MP with the aim of assessing the mobility of local protein, as represented in Figure 2B. The depicted plot was generated against residue number along the course of a 30 ns trajectory. The resulting RMSF profile obtained displayed fluctuations observed at the catalytic site of P450, spanning a range of approximately 0.05 to 0.42 nm for both systems. Interestingly, the RMSF value of the unbound enzyme is generally higher than that of the 8-MP-CYP1B1 complex, suggesting the constraining capability of the drug to the enzyme fluctuations. Additionally, no notable fluctuations were detected at the 8-MP binding site in the 8-MP-CYB1B1 complex compared to the unbound protein. Also, the outputs of the RMSF profile indicate that fluctuations exceeding 0.4 nm are associated with residues distant from the 8-MP-binding pocket. Moreover, the residue in contact with the 8-MP exhibits the highest stability, reflected in its low RMSF value. Hence, the data obtained clearly support the conclusion that residues within the main binding site exhibit minimal fluctuations concerning 8-MP, suggesting that the architecture of the 8-MP binding site remains largely rigid throughout the 30 ns MD simulations.

The temporal changes in the radius of gyration (Rg) provide valuable insights into the dynamics of protein collapse. The Rg values elucidate the compactness of the enzyme under investigation, reflecting its folding and unfolding dynamics throughout the 30 ns MD simulations based on thermodynamic principles [11]. Figure 2C represents the calculated Rg values of free CYB1B1 and 8-MP complexed with CYB1B1 as a function of time. Obviously, both systems exhibited constant values at about 7.5 ns, indicating that the enzyme and 8-MP complex reached equilibrium at about 7.5 ns of the simulation. The compactness of the CYB1B1 geometry along the simulation was assessed by observing the decrease in the Rg values of the backbone atoms. The average Rg value of the enzyme and the drug–enzyme complex is 2.25 ± 0.07 nm. These findings are in agreement with prior MD studies on P450 enzymes, indicating that drug binding decreases protein densification and enhances its structural compactness [11].

Frequently, SASA acts as a parameter to characterize the ratio of the enzyme–solvent interactions, which can predict the extent of conformational variations during the binding mechanism and assess protein accessibility [24]. The SASA changes in both studied systems during the 30 ns MD simulation are represented in Figure 2D. The outputs of SASA calculations of unbound CYP1B1 and 8-MP-P450 complex revealed a normal fluctuation within the range of approximately 210–225 nm^2^, and both systems attained equilibrium directly with the start of the simulation. The results of SASA calculations are complementary to that of Rg values; this correspondence underscores the validity and precision of the outcomes derived from the MD simulations.

Figure 2E represents the Coulomb (Coul) and Lennard–Jones (LJ) interaction energies (GROMACS energies) between CYB1B1 and 8-MP during the simulation; these energies were regarded as short-range (SR) energy types extracted from the MD simulation trajectory files. The Coul-SR energy is a good predictor of the drug–enzyme complex equilibrium along the MD simulation duration. In this intricate system, minimal fluctuations were observed in the Coul-SR interaction energy, with only one instance of a notable energy decrease identified within the temporal range of approximately 17–19 ns. The Coul-SR GROMACS energy for the studied complex ranged between −50 and 15 kJ/mol, and the running average Coul-SR energy is −16 ± 4 kJ/mol. On the other hand, the LJ-SR interaction energy of the complex revealed high fluctuation at MD run initialization and then attained equilibrium at about 6 ns. The 8-MP-CYP1B1 LJ-SR energy values ranged approximately from −70 to −130 kJ/mol and were reasonably stable. The LJ-SR energies were deemed rational indicators for predicting binding interactions. Thus, the compatibility of 8-MP to the binding site of CYP1B1 is mainly attributed to the energetically favorable interactions as estimated by Coul-SR LJ-SR interaction energy calculations.

The binding nature of 8-MP within the binding cavity of CYP1B1 was explored by assessing the hydrogen bonding profile of the drug–enzyme complex along the 30 ns MD run, as shown in Figure 2F. Intriguingly, 8-MP displayed a weak hydrogen bonding pattern with the active site amino acid residues. This insight is particularly intriguing as it illuminates the hydrophobic characteristics of the CYP1B1 binding site, as predicted by molecular docking analyses. This outcome will be further utilized in studying the reactive metabolite formation mechanisms.

### 2.3. A Simplified Model for 8-MP-P450

In order to establish a streamlined process for our DFT examination, we formulated a simplified model that encompasses the various proposed pathways for the MBI of P450 by 8-MP, as shown in Figure 3. In this model, we used the Fe^IV^ = O moiety ingrained into a doubly protonated porphyrin ring to represent the Cpd I active site of P450. The axial proximal ligand was symbolized as HS−, as previously reported [25]. This model is reputed to precisely mimic the electronic structures and high reactivity of P450s, thereby making it a trustworthy tool for exploring the molecular mechanisms of P450-catalyzed biotransformation [26]. Nevertheless, this model does not consider actual porphyrin side chains and P450 active site amino acid residues for their potential influences on drug binding and activation, which could pose challenges in accounting for enzymatic selectivity and inevitably lead to barrier error. However, it is worth noting that prior studies demonstrated that the use of different cysteine axial ligand substitutions yields similar energy profiles [25,27]. Furthermore, in quantum chemical models, the incorporation of porphyrin side chains typically results in only a slight increase in the activation energies of 1–2 kcal mol^−1^ for the hydrogen abstraction transition states [25,27].

In Figure 1, we present the possible pathways for the MBI of P450 by 8-MP, which are differentiated by the interaction site in 8-MP susceptible to P450 binding (highlighted in Figure 3). The potential biotransformation routes include *O*-demethylation of the terminal methyl in C-8 (Path A), epoxidation of the double bond at C3-C4 (Path B), and epoxidation of the double bond at C4′-C5′ (Path C). We also explored the directing energetic influence of phenylalanine residues on the epoxidation pathway in Path C. All proposed biotransformation routes were investigated from an energetic perspective in both doublet and quartet states. Additionally, the reactive metabolites derived from epoxidation products in Path C were also examined further, as depicted in Figure 2. Figure 4, Figure 5 and Figure 6 present the calculated energy profiles (in kcal mol^−1^) and key geometrical parameters for Paths A, B, and C at the UB3LYP/B2//B1+ZPE (B1) and UB3LYP (SCRF)/B2//B1+ZPE (B1) levels for both the high spin (HS) and low spin (LS) states of the MBI of P450 by 8-MP at the potential interaction sites in the 8-MP structure.

### 2.4. O-Demethylation Pathway (Path A)

Among the various reactions catalyzed by cytochrome P450 enzymes, *O*-demethylation is a significant metabolic pathway involved in the biotransformation of numerous drugs, environmental pollutants, and natural compounds [28]. *O*-demethylation catalyzed by P450 entails the removal of a methyl group from a P450-hydroxylated substrate, typically resulting in the formation of a hydroxyl group [29]. This enzymatic process is pivotal in drug metabolism, as it often leads to the generation of active metabolites or facilitates the detoxification of xenobiotics [29]. However, this metabolic pathway is initiated by the C-H hydroxylation mechanism. Typically, C-H hydroxylation mechanisms initiated by P450 begin with a hydrogen atom transfer from the substrate to the oxo moiety of the P450 compound I (Cpd I) [11]. This is typically the rate-determining step for the majority of C-H hydroxylation mechanisms [30]. Following this, a ferryl hydroxyl porphyrin is formed as an intermediate, with the carbon radical still coordinated to the hydroxyl group of the P450 active site (Cpd I) [30]. Upon formation of the radical intermediate, a rebound of OH from P450 to the hydroxylation site of the substrate leads to the creation of a hydroxylated substrate [11]. 8-MP could undergo C-H hydroxylation at the terminal methyl group at C-8, facilitating the initiation of the MBI of P450 (Figure 3, Figure 1). The energy profile for this metabolic pathway is shown in Figure 4. Initially, the reactant complex is stabilized by a hydrogen bond between the oxo moiety of Cpd I and the methyl C-H of 8-MP (O...H distance is 2.36 Å for both spin states). The activation energies for the initial H-abstraction are 12.60 and 13.91 kcal mol^−1^ for doublet and quartet states, respectively. The derived hydrogen transfer transition states exhibit the typical characteristics of H-abstraction transition states [11]. The formation of the radical intermediate (^2,4^INT-H_A_) exhibits an endothermic character, with the ground state energies for both HS and LS states being higher than their preceding reactant complex. The newly formed O...^•^C bond is weak (3.28 and 3.22 Å for LS and HS, respectively), facilitating the rebound of OH to the alkyl radical. The doublet-state rebound process proceeds without a rebound barrier. This outcome, along with the lower LS H-abstraction energy barrier, suggests that this pathway is more likely to proceed via a doublet spin state. These results align with previous findings regarding the hydroxylation of various substrates [28,29]. The energy of the hydroxylated 8-MP (^2,4^P-OH_A_) in both doublet and quartet states is lower than that of ^2,4^RC_A_ by approximately 50 and 60 kcal mol^−1^, respectively.

### 2.5. Epoxidation Pathways (Paths B and C)

Epoxides are pivotal intermediates in both organic and biosynthetic processes, capable of undergoing a variety of reactions, including nucleophilic substitution and hydrolysis. These reactions often yield a wide array of products [23]. Epoxidation processes involving metal complexes, such as iron-containing metalloporphyrin, are not frequently studied in quantum chemical investigations [31]. Nevertheless, discussions of drug epoxidation reactions involving an oxidant model that simulates the Cpd I active site of P450 do exist [23], including examples like the epoxidation of ethene, propene, 1-butene, and trans-2-butene [32,33]. The epoxidation mechanism generally proceeds in two steps: an initial O attack on the target C=C bond to form a C-O bond, followed by a ring closure step that leads to the formation of the epoxidated substrate (Figure 1) [34]. Two olefinic double bonds in 8-MP, C3=C4 (Path B) and C4′=C5′ (Path C) are susceptible to epoxidation by P450 (Figure 1). The energy profiles for these proposed epoxidation pathways, estimated via our quantum chemical calculations, are shown in Figure 5 and Figure 6. The estimated oxygen addition energy barriers (^2,4^TS-O_B_) for Path B are 16.03 and 14.32 kcal mol^−1^ for both HS and LS states, respectively. Conversely, relatively lower rate-determining state energy barriers (^2,4^TS-O_C_) were obtained for Path C (13.27 and 13.06 kcal mol^−1^ for quarter and doublet states, respectively). This finding is particularly intriguing as it suggests a thermodynamic preference for Path C as the prevailing route for the catalyzed biotransformation of 8-MP by P450. In Path C, the effect of solvation (*ε* = 5.7) led to a slight increase in the transition state energies. This increase is primarily attributed to the enhanced localization caused by the medium polarization effects of the employed solvent (chlorobenzene) [35]. Various bond distance variations during both mechanisms (Path B and C) align with previously reported bond distance variations for the epoxidation mechanism by P450 [23]. Earlier data on the epoxidation of furan revealed a quite negligible rebound barrier for the quartet state and a barrier-free mode for the doublet spin state [23]. In contrast, the ring closure step in our mechanisms proceeds via barrierless routes in both doublet and quartet states. Moreover, the estimated product complexes (^2,4^P-epo) in both mechanisms exhibit weaker Fe…O distances and are significantly more stable than their preceding reactant complex, particularly in the doublet spin state by approximately 30 kcal mol^−1^.

The energetic coexistence of the three mechanisms, namely Paths A, B, and C, in the potential metabolic transformation of 8-MP by P450, can be inferred from a thermodynamic perspective. This is based on the favorable H-abstraction transition state, the stabilities of hydroxylated 8-MP, and the small rebound barrier observed in Path A. Additionally, Path B does not deviate greatly from the energy requirements of Path C, thus suggesting that Path C could be the kinetically possible route. However, it is crucial to note that the epoxidation of the five-membered ring in Path C yields a metabolic product that is susceptible to further transformations such as ring-opening, nucleophilic addition, or interactions with key amino acid residues [34]. Consequently, the resulting epoxide, ring-opened products, or the formed reactive metabolites could all play significant roles in inactivating P450 through various covalent bindings to the enzyme’s active site. In light of these observations, further investigations are warranted to uncover the different mechanisms that underlie the formation of reactive metabolites from the epoxidation products in Path C. This will allow for a more comprehensive understanding of the potential metabolic transformations of 8-MP by P450, thus providing more information that could be valuable in the context of pharmacology and drug design.

### 2.6. Reactions of 8-MP Epoxide Reactive Metabolites

The formation of reactive metabolites during catalyzed biotransformation reactions is a prevalent phenomenon. As outlined in Figure 2, several proposed mechanisms exist for the formation of reactive metabolites in the metabolism-based inactivation (MBI) of P450 by 8-MP. One common route, supported by numerous discussions, is the nucleophilic attack by key amino acid residues in the enzyme’s binding site [36]. This interaction can lead to the formation of a covalent adduct, which may block the catalytic site of P450, thus enabling efficient MBI of P450 (Figure 2a,b) [23]. In our study, we selected phenylalanine residue Phe268 for further investigation due to its significant contribution to the binding profile of 8-MP with CYP1B1 (Figure 1). This choice was supported by molecular docking results and previous reports indicating the thermodynamic favorability of this residue’s interaction in the binding modes of CYP1B1 with various substrates [6]. A second major pathway involves a non-catalyzed ring cleavage reaction that generates an electrophilic dicarbonyl intermediate (Figure 2d). This route was implicated in the metabolic mutagenicity and carcinogenicity of numerous substrates [37]. A third notable pathway is the hydrolysis of the 8-MP epoxide, driven by the cellular environment’s bulk polarity, to form a vicinal diol. This diol can undergo further ring-opening reactions to yield adverse reactive metabolites (Figure 2e). The energy profiles of these various reactive metabolite formation pathways, along with the key geometrical parameters of transition states, are depicted in chlorobenzene solvent in Figure 7. These illustrations serve to provide a more detailed picture of how these mechanisms might proceed and their potential implications on the inactivation of P450 enzymes.

The formation of covalent adducts via nucleophilic amino acid residues in the active site of P450 is an established route for the effective MBI of P450 [36]. These residues typically display an anionic character, especially when interacting in a physiological environment [23]. Given the electrophilic nature of the formed 8-MP epoxide, it is a likely target for attack by the active OH and SH sites in nucleophilic amino acids, which are often initiated by the effect of lone pairs of electrons on sulfur and oxygen atoms. In biological interactions, terminal OH and SH sites on interacting with nucleophilic amino acids are prone to ionize to -OR and -SR, respectively, facilitated by basic residues and reaction conditions. Thus, we evaluated the nucleophilic addition reactions of anionic moieties -OMe and -SMe (as representatives of nucleophilic amino acids) to 8-MP epoxide, as depicted in Figure 2a,b. Neutral nucleophilic addition of ^−^OMe and ^−^SMe was considered on the electron-deficient C-5′ of the formed 8-MP epoxide, leading to the formation of P_OMe_ and P_SMe_, respectively (Figure 2a,b). Our DFT calculations revealed energy barriers of 29.53 and 33.68 kcal mol^−1^ for the transition states (TS_OMe_ and TS_SMe_) for ^−^OMe and ^−^SMe anionic additions, respectively (Figure 7a,b). Notably, these activation energy barriers are substantially lower than those previously calculated for the neutral nucleophilic addition to furan epoxide (40.27 kcal mol^−1^) [23]. Additionally, the resulting covalent adducts in both scenarios are more stable than their preceding reactant complexes. However, these nucleophilic adducts are likely to undergo ring opening to proceed along their catalyzed biotransformation pathways. This information leads us to propose that these neutral nucleophilic additions may contribute to potential P450 MBI via an acid–base catalysis approach.

Our docking studies identified three phenylalanine residues within the binding pocket of the 8-MP-CYP1B1 complex. These residues were suggested to exhibit energetically favorable interactions in the binding configuration of this complex [6]. As a result, we sought to explore the role of the abundant phenylalanine residues in the binding mechanism of this metabolic process. We used DFT analysis to investigate the neutral nucleophilic addition of phenylalanine to the formed 8-MP epoxide, as depicted in Figure 2c. The energy profile for this mechanism is illustrated in Figure 7c. Remarkably, the activation energy barrier for this reaction (TS_phe_) is lower than those derived for the neutral OMe and SMe additions, measuring at 25.71 kcal mol^−1^. However, the resulting phenylalanine adduct (P_Phe_) is somewhat less stable, suggesting the likelihood of additional ring-opening reactions to complete its biotransformation pathway. Given these findings, phenylalanine may play a key role as a nucleophilic residue contributing to the addition to the 8-MP epoxide.

Autocatalyzed rearrangement of substrate epoxides to produce electrophilic species is not a commonly observed metabolic pathway, as illustrated in Figure 2d. This mechanism is primarily driven by the influence of the lone pair of electrons on the furanyl O atom, enabling a noncatalyzed ring-opening reaction. One such proposed metabolic route for 8-MP epoxide to complete the biotransformation process is through a non-enzymatic direct ring cleavage. It has been reported that furan epoxides can undergo autocatalytic ring-opening events, resulting in the formation of electrophilic intermediates like cis-2-butene-1,4-dial [23]. These intermediates can then undergo nucleophilic addition by compounds like glutathione (GSH), DNA, or amino acid residues [23]. The resulting dialdehyde addition products often produce harmful carcinogenic metabolites [23]. The energy profile for the non-CYP-catalyzed 8-MP epoxide ring opening reaction is depicted in Figure 7d. The calculated activation energy barrier (TS_ro_) for this pathway is lower than those obtained for all proposed reactive metabolite formation mechanisms (13.79 kcal mol^−1^). However, this reaction is highly endothermic, yielding a product (P_ro_) energy of 7.69 kcal mol^−1^, which is higher than the initial reactant epoxide. This product may undergo further noncatalytic degradation, leading to the generation of potentially harmful reactive metabolites. Due to the lower energy requirements for this pathway, it is reasonable to consider this route as one of the main contributing pathways in the metabolism of 8-MP. There is evidence suggesting that long-term oral administration of 8-MP combined with UVA therapy may increase the risk of non-melanoma skin cancer [38]. Moreover, topical application of this treatment has been reported to induce malignant tumors in experimental animals [38]. Therefore, it is plausible to associate the nonenzymatic ring opening reaction as one potential pathway contributing to the undesirable mutagenicity of 8-MP. These findings guide us to further investigate this route for understanding the mutagenicity and carcinogenicity associated with the use of 8-MP in the treatment of certain skin conditions.

Another potential metabolic route for 8-MP epoxide is hydrolysis in the active site of P450, driven by the inherent polarity of the protein environment, which could produce a vicinal diol as illustrated in Figure 2e. However, our docking and MD simulation assessments indicate a hydrophobic nature for the binding of 8-MP to the active site of CYP1B1, making the hydrolysis of the epoxide unlikely due to the hydrophobic character of the binding pocket. Nevertheless, we used DFT calculations to explore the mechanistic details associated with the nucleophilic addition of water to the formed epoxide. This metabolic pathway is suggested to proceed through an initial attack by oxygen on C-5′, producing a vicinal diol. This diol could then undergo one of two possible ring-opening routes to generate reactive metabolites. The energy profile for the hydrolysis of the epoxide followed by ring cleavage is depicted in Figure 7e. A relatively high energy barrier was obtained for the formation of the diol (31 kcal mol^−1^). The resulting diol could potentially undergo ring-opening via C4′-C5′ bond cleavage or C5′-O bond cleavage. The transition state for the C-C bond cleavage is thermodynamically disfavored due to the high energy requirement for this rate-determining step (51.71 kcal mol^−1^). In contrast, the ring opening pathway through C-O bond cleavage is energetically more feasible (21.90 kcal mol^−1^). However, the energy barrier for diol formation is the highest among the barriers for the various initial transition states in this enzymatic system. Therefore, it is unlikely that the diol formation process significantly contributes to the metabolic inactivation of P450 by 8-MP. Considering the significant energy barrier variations among all studied mechanisms, we propose that the nucleophilic addition, especially by phenylalanine, to the 8-MP epoxide is the most plausible mechanism for the formation of reactive metabolites. The covalent adduct obtained through nucleophilic addition is likely to play a significant role in the metabolic inactivation of P450 by 8-MP.

### 2.7. The Driving Effect of Phe268

Numerous studies have highlighted the vital catalytic role of phenylalanine residues, particularly Phe134, Phe231, and Phe268, in the binding profile of CYP1B1 with various inhibitors [6,10,39]. These residues have been shown to exhibit energetically favored π–π stacking interactions with many substrates and P450 inhibitors [10]. Our docking and MD simulation analysis results revealed the hydrophobic interactions of these residues with 8-MP in the active site of CYP1B1. Our DFT analysis suggests that the prevailing route for the formation of reactive metabolites that inactivate P450 by forming a covalent adduct is the epoxidation of the five-membered ring of 8-MP. The residue Phe268 was found to be in direct hydrophobic contact with the active C=C bond in 8-MP (Figure 1). Thus, we decided to examine the directing effect of Phe268 on the metabolic epoxidation pathway of 8-MP through DFT analysis (Figure 1). The energy profile for the epoxidation of the C4′=C5′ double bond in 8-MP, catalyzed by P450 and phenylalanine, is presented in Figure 8. Initially, the reactant complex (^2,4^RC_ph_) is stabilized by strong hydrogen bonding between the OH of phenylalanine and the O atom of Cpd I (1.66 Å). Simultaneously, the catalytic residue secures the substrate with two polar interactions: one between the NH of phenylalanine and the OMe group of 8-MP (2.30 Å), and another between the methyl group of 8-MP and the carbonyl oxygen of the amino acid (2.47 Å). A weak polar interaction was also observed between the carbonyl oxygen of phenylalanine and a hydrogen atom in the porphyrin ring. Next, the oxygen addition transition states (^2,4^TS-O_ph_) displayed the same bonding pattern as the reactant complex, with minor modifications, such as a decrease in the bond distance between the O of Cpd I and the CH of 8-MP, as well as the bond between the carbonyl O and Me of the substrate. The formed intermediates (^2,4^INT_ph_) and the epoxide product (^2,4^P-epo) presented different binding profiles with variable bond distances as displayed in Figure 8. In these profiles, the catalytic residues formed one polar bond between its OH group and a carbon atom from the porphyrin ring and another bond between the carbonyl oxygen and the methyl group of 8-MP.

The catalytic role of phenylalanine in the binding mechanism of this metabolic process appears to contribute to a slight reduction in the activation energy barrier of the transition state (12.14 and 11.39 kcal mol^−1^ versus 13.06 and 13.27 kcal mol^−1^ for both the low spin (LS) and high spin (HS) state, respectively). Moreover, the catalytic influence of phenylalanine results in a substantial increase in the stability of the obtained epoxide in the quartet spin state (−17.77 kcal mol^−1^ versus −28.54 kcal mol^−1^). These findings suggest that Phe268 plays an energetically favorable role in the binding mechanism of 8-MP with CYP1B1. Furthermore, the extensive hydrogen bonding network exerted by the presence of phenylalanine seems to play a significant role in retaining the substrate within the active site of P450.

## 3. Experimental

### 3.1. Molecular Docking Analysis

Autodock Tools (ADT) v1.5.6 and AutoDock Vina v 1.1.2 software were used to carry out molecular docking studies for the interactions of 8-MP with CYP1B1 [40]. The crystallized three-dimensional structure of P450 1B1 isoforms was obtained from the Protein Data Bank (PDB). The pdb id of CYP1B1 is 3PM0. The geometrical structure of 8-MP was fully optimized at the B3LYP level [41,42,43] with the 6-311G (d, p) basis set [44] using the Gaussian 09 software package [45]. 8-MP was docked into the main active site of CYP1B1. The initial structure of CYP1B1 was visualized and deprived from native ligands and solvent molecule residues by using UCSF Chimera [46]. The selected isozyme was optimized for docking by using ADT. This optimization process comprises the addition of polar hydrogens and setting the grid box to the original ligand binding pocket. PyMOL v2.3.2. was used for examining the binding site, molecular visualization, and high-resolution image generation.

### 3.2. Molecular Dynamics Simulations

The PDB file of the 8-MP-CYP1B1 complex with the lowest binding energy generated by molecular docking was employed as the initial geometrical structure for executing MD simulations. The 30 ns MD simulations of CYP1B1 in water and 8-MP-CYP1B1 complex were performed by means of The GROMACS 2022.4 software package [47,48]. The structure-balanced CHARMM36m force-field was employed for all-atoms MD run of CYP1B1 and 8-MP-CYP1B1 complex [49]. The CGenFF served was used to generate the topology and geometrical parameters of 8-MP (https://cgenff.umaryland.edu/ accessed on 2 January 2024). The complex and the enzyme were placed in a dodecahedron box with the periodic boundary conditions; the new box volume is 530.48 nm^3^. For solvation, the CHARMM-modified TIP3P water model was employed [50]. Both studied systems were electro-neutralized by adding three chloride counter-ions.

The steepest descent energy minimization technique was employed for a duration of 10 ps to alleviate unfavorable thermodynamic interactions [51]. Next, both systems underwent two stages of equilibration at 300 K (nsteps = 50,000), using NVT and NPT ensembles [52]. Subsequently, a 30 ns MD simulation run was conducted at 300 K temperature and 1 bar pressure. The MD simulation protocol was adopted from earlier works documented elsewhere [11,53].

### 3.3. DFT Methodology

In this study, we used oxo-ferryl porphyrin (Fe^4+^O^2−^(C_20_N_4_H_12_)^−^(SH)^−^) as a simplified model for the compound I (Cpd I) form of the active site of P450 [11,12,30]. Gaussian 09 software package was employed to execute all DFT calculations for the proposed mechanisms of 8-MP-P450 interaction [45]. All proposed mechanism-constituting structures in both high spin (HS) and low spin (LS) states underwent geometry optimizations and frequency calculations using the unrestricted B3LYP [41,42,43,54] exchange–correlation functional along with the double-ζ LANL2DZ basis set for the iron atom and 6-31G* basis set for the remaining atoms (symbolized BS1). The outcomes of these caclculations allow for the elucidation of the stationary points, identify zero-point energy (ZPE), and compute the imaginary frequencies of various transition states. Then, single-point energy calculations were performed with the LANL2DZ (F) (Fe)/6-311+G** (H, C, N, O, S) basis set (symbolized BS2) in the gas phase (*ε* = 1) and in chlorobenzene solvent (*ε* = 5.7) to simulate the bulk polarity effects of the cellular protein [55,56,57]. The polarizable continuum model (PCM) solvation model was employed to estimate the solvation influences using the self-consistent reaction field (SCRF) method [58,59]. The reader is encouraged to refer to the Appendix A for detailed information on the energy and coordination of the structures addressed in the current study.

## 4. Conclusions

To sum up, we conducted a comprehensive computational exploration to elucidate the mechanistic aspects of 8-MP inhibitory activity against P450 enzymes within the context of mechanism-based inactivation (MBI). We examined three potential inactivation pathways for P450 by 8-MP, namely C-H hydroxylation of 8-OMe, C3=C4 epoxidation, and C4′=C5′ epoxidation. These paths encompass all possible interaction sites in 8-MP that could partake in metabolic processes. Comparative study results suggest energetic and condition favorability for the epoxidation of the 8-MP five-membered ring, but the methyl hydroxylation route remains a plausible contributor to the metabolic mechanism due to similar energy demands.

We further explored the potential formation of reactive metabolites from the predominant thermodynamic route’s ensuing metabolic processes. The 8-MP epoxide may undergo a nucleophilic attack by key interacting residues, non-CYP catalyzed ring cleavage, or hydrolysis to yield a diol, followed by ring opening, resulting in reactive metabolites. Our DFT analysis illuminated the likelihood of forming covalent nucleophilic epoxide adducts through key amino acid residue addition and reactive metabolite formation due to non-enzymatic catalyzed ring cleavage of 8-MP epoxide. Therefore, the critical factors promoting P450 MBI by 8-MP include the following: (a) the abundance of nucleophilic amino acid residues in the active site interacting directly with the substrate; (b) the hydrophobic environment in the binding site, mitigating the competition from the diol formation pathway; (c) the constant involvement of the heme moiety throughout the catalyzed biotransformation process, making it more susceptible to inactivation.

Finally, we evaluated the impact of the key residue, Phe268, on the metabolic pathway of epoxide formation. Phenylalanine showed multiple hydrogen bonding interactions with the substrate and with Cpd I of P450, playing a critical role in sequestering 8-MP within the P450 binding site. The catalytic role of phenylalanine significantly reduced the energy requirements of the transition state and enhanced the stability of the formed high-spin (HS) epoxide. As a result, 8-MP, bolstered by phenylalanine catalysis, appears to be a potent P450 inactivator, acknowledging the potential for adverse reactive metabolite formation. This study offers molecular-level insights into P450 MBI by 8-MP epoxide reactive metabolites. Such understanding can assist in early drug discovery stages by illuminating potential risks.

## Data Availability

The data are available in Appendix A.

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
