# Peer review of "Deciphering the Molecular Mechanisms of Reactive Metabolite Formation in the Mechanism-Based Inactivation of Cytochrome p450 1B1 by 8-Methoxypsoralen and Assessing the Driving Effect of phe268"

_molecules, 2024, doi:10.3390/molecules29071433_

Round 1

Reviewer 1 Report

Comments and Suggestions for Authors

This study, which is devoted to computational analysis of metabolism-based inactivation of CYP1B1 by 8-methylpsoralene, is well written, adequately designed and performed, and is of interest to the P450 community. I have only one substantial concern: when outlining the potential mechanisms of metabolism of methoxypsoralen and possible pathways for the mechanism-based inactivation, the authors list three possible P450-catalyzed pathways: two epoxidation reactions and hydroxylation of methyl carbon in the methoxy group in the 8-th position. However, the latter reaction is very exotic for P450 enzymes: in most cases, the P450 attack on a methoxy or methylamino group results in demethylation and formation of formaldehyde. The intermediate with hydroxylated methoxy- or methylamino group is unstable and decomposes with a release of formaldehyde  - see https://doi.org/10.1021/tx0002583 (section 3.2.3. and Scheme 9 there). These oxidative demethylation reactions are very common in P450 catalysis and take place with myriad substrates. Thus, the authors may need to replace the pathway of methoxy group hydroxylation with an oxidative demethylation pathway and modify the discussion accordingly.   

Besides this major concern, I have several minor corrections to the text:

-          Line 68: An abbreviation “MBI” for mechanism-based inactivation appears on this line for the first time. However, its definition is given only on line 76. You should move it to the place of the first use.

-          Line 130: “isolated from native ligands, solvents, and nonstandard residues” sounds bizarre. A better choice is “deprived of native ligands, solvents, and nonstandard residues”. Also, it is not clear which “nonstandard residues” you mean here. Are there any?

-          Line 261: “within the range of approximately 210-225 nm” – I presume that the authors mean “210-225 nm2” (nanometers-square, the surface unit)

-          Lines 290-291: “only one trough was detected within the range of approximately 17-19 nm” – I do not understand which “trough” you mean here and what nanometers (or nanometers-square??) are doing in the discussion of the Coulomb and Lennard-Jones interaction energies. Please, explain better or delete the phrase.

Reviewer 2 Report

Comments and Suggestions for Authors

This work is a thorough examination into the activity 8-MP on P450. The work includes DFT calculations, MD and docking. In my opinion, the work provides useful insights and deserves to be published. Some points can be expanded to provide further clarification.

For the MD simulations, only the lowest binding energy generated by molecular docking was employed. What about other docking poses? Is it safe to discards in this case?

The authors center the DFT discussion in a detail energetic analysis. Further insight can be obtained if electronic descriptors are used in the discussion of the epoxidation and hydroxylation pathways.

The description of open shell systems presents a challenge for DFT. What is the error in the estimated activation energies?

Round 2

Reviewer 1 Report

Comments and Suggestions for Authors

The authors have successfully eliminated all the shortcomings. The manuscript may be published in its current form.